# Genomic Characterization of Non-Invasive Differentiated-Type Gastric Cancer in the Japanese Population

**DOI:** 10.3390/cancers12020510

**Published:** 2020-02-22

**Authors:** Koki Nakamura, Yuji Urabe, Kenichi Kagemoto, Ryo Yuge, Ryohei Hayashi, Atsushi Ono, C. Nelson Hayes, Shiro Oka, Masanori Ito, Takashi Nishisaka, Kazuaki Tanabe, Koji Arihiro, Hideki Ohdan, Shinji Tanaka, Kazuaki Chayama

**Affiliations:** 1Department of Gastroenterology and Metabolism, Institute of Biomedical and Health Science, Hiroshima University, 1-2-3 Kasumi, Minami-ku, Hiroshima 734-8551, Japan; nkoki@hiroshima-u.ac.jp (K.N.); beyan13@hiroshima-u.ac.jp (Y.U.); ken09090303@hotmail.co.jp (K.K.); atsushi-o@hiroshima-u.ac.jp (A.O.); nelsonhayes@hiroshima-u.ac.jp (C.N.H.); oka4683@hiroshima-u.ac.jp (S.O.); maito@hiroshima-u.ac.jp (M.I.); 2Department of Regeneration and Medicine Medical center for Translation and Clinical Research, Hiroshima University Hospital, Hiroshima 734-8551, Japan; 3Department of Endoscopy, Hiroshima University Hospital, Hiroshima 734-8551, Japan; makapoo@hiroshima-u.ac.jp (R.Y.); r-hayashi@hiroshima-u.ac.jp (R.H.); colon@hiroshima-u.ac.jp (S.T.); 4Department of Pathology and Laboratory Medicine, Hiroshima Prefectural Hospital, Hiroshima 734-8551, Japan; tnishisaka@hph.pref.hiroshima.jp; 5Department of Health Care for Adults, Graduate School of Biochemical & Health Sciences, Hiroshima 734-8551, Japan; ktanabe2@hiroshima-u.ac.jp; 6Department of Anatomical Pathology, Hiroshima University Hospital, Hiroshima, Hiroshima 734-8551, Japan; arihiro@hiroshima-u.ac.jp; 7Department of Gastroenterological and Transplant Surgery, Graduate School of Biochemical & Health Sciences, Hiroshima 734-8551, Japan; hohdan@hiroshima-u.ac.jp

**Keywords:** gastric cancer, driver mutation, LRP1, CagA, *Helicobacter pylori*

## Abstract

Background and aims: Recent genomic characterization of gastric cancer (GC) by sequencing has revealed a large number of cancer-related genes. Research to characterize the genomic landscape of cancer has focused on established invasive cancer to develop biomarkers for therapeutic or diagnostic targets, and nearly all GC reports have been about advanced GC. The aim of this study is to identify recurrently mutated genes in non-invasive GC and, in particular, the driver mutations that are associated with the development of GC. Methods and results: We performed whole-exome sequencing of 19 fresh frozen specimens of differentiated-type non-invasive GC and targeted sequencing for 168 genes of 30 formalin-fixed paraffin-embedded archival specimens of differentiated-type non-invasive GC. We found that TP53 and LRP1 are significantly associated with non-invasive GC. It has been reported that LPR1 is associated with CagA autophagy in gastric mucosa. Therefore, we downloaded RNA sequence data for gastric cancer from the The Cancer Genome Atlas (TCGA) Genomic Data Commons Data Portal and examined the differences in LRP1 gene expression levels. The expression level was significantly lower in cases without LRP1 mutation than in cases with LRP1 mutation. Based on these results, fluorescent immunostaining for CagA was performed for 49 of the above samples to evaluate CagA accumulation within the cancerous tissue. Accumulation of CagA was significantly greater when an LRP1 mutation was present than without a mutation. Conclusion: These data suggest that LRP1 mutation is an important change promoting the transformation of gastric mucosa to GC early in the carcinogenesis of cancer.

## 1. Introduction

Gastric cancer (GC) is the third most common cause of cancer-related mortality in the world [1]. The highest incidences of GC are in East Asia, Central and Eastern Europe, and South Africa [2]. In Japan, more than 130,000 people suffer from GC, which is ranked as the 3rd most common cause of death due to carcinoma. The majority risk factor for GC is *Helicobacter pylori* (*Hp*) infection [3]. The rate of *Hp* infection in Japan is approximately 80% in individuals over age 50, and about 20% in individuals younger than age 20. Moreover, 0.4% of patients with *Hp* infection develop GC each year in Japan. Based on epidemiologic evidence, high intake of nitrosamines, processed meat products, salt and salted foods, obesity, and smoking are all associated with an increased risk of GC [4]. Furthermore, some recent studies have reported that Epstein-Barr Virus infection increases the risk of GC [5]. The most prevalent form of GC begins with chronic gastritis due to *Hp* infection, which leads to atrophic gastritis and intestinal metaplasia and, finally, to dysplasia and adenocarcinoma [6]. Some reports have noted that *Hp* infection promotes genetic alterations in the gastric epithelium [7,8].

Advances in endoscopic technology have made it possible to detect GC at an early stage, which allows for prompt administration of further endoscopic treatment, such as endoscopic mucosal resection or endoscopic submucosal dissection (ESD). Recent genomic characterizations of GC have identified a large number of cancer-related genes. Research to characterize the genomic landscape of cancer has focused on established invasive cancer to develop biomarkers for therapeutic or diagnostic targets [9]. However, it is becoming clear that extensive genomic heterogeneity is present in the majority of advanced cancers [10]. Using whole-genome sequencing of 100 GC samples, Wang et al. identified TP53, ARID1A, CDH1, MUC6, CTNNA2, GLI3, RNF43 as GC-related driver genes [11]. The Cancer Genome Atlas (TCGA) project proposed to divide GC into four subtypes: Epstein-Barr Virus (EBV)-positive, characterized by recurrent PIK3CA mutations and DNA hypermethylation; microsatellite instability (MSI) tumors; genomically stable tumors, which display recurrent RHOA mutations; and tumors with chromosomal instability, which feature recurrent TP53 mutations and focal amplification of receptor tyrosine kinases [12]. However, almost all of these reports on GC relate to advanced GC, and few comprehensive approaches have been performed with respect to non-invasive GC. Here, we performed sequencing in non-invasive GC and searched for recurrently mutated genes.

## 2. Results

### 2.1. Subsection

#### 2.1.1. Study Design and Patient Characteristics

This study design and patient characteristics are shown in Figure 1 and Appendix A. 

First, we performed whole-exome sequencing in the initial screening stage. We used FF samples of 19 patients (14 male, five female; mean age: 74 years) who consented to provide GC samples. All lesions had been detected by endoscopic screening for gastro-intestinal carcinoma or by endoscopic surveillance after treatment for gastro-intestinal disease. Each GC was classified as a differentiated intramucosal carcinoma.

Second, we performed targeted sequencing in the replication stage. We used formalin-fixed paraffin-embedded (FFPE) samples of 30 additional patients (25 male, five female; mean age: 72 years) who consented to provide GC samples. We selected GCs that are differentiated intramucosal carcinomas.

Third, we performed targeted sequencing to verify whether the data obtained in the replication stage is applicable even in the case of advanced GC. We used FF samples of 19 advanced GC patients (16 male, three female; mean age 71 years).

#### 2.1.2. In the Screening Stage

First, we performed whole-exome sequencing on 19 matched gDNA samples from non-invasive differentiated-type GCs and peripheral blood leukocytes. All samples were obtained by biopsy just before ESD. We marked the spot of biopsy at ESD and estimated the tissues under pathological review. Then we performed Sanger sequencing to confirm the identified mutations. We identified 134 mutations using a random approach in each sample, and all 134 mutations in 19 samples were successfully validated by Sanger sequencing (Appendix A). The mean read depth was 105-fold (Appendix A), and 97.95% of target bases were covered by >10 independent readings (Appendix A). A total of 3947 somatic events, including 3227 single-nucleotide substitutions and 720 short insertions and deletions (indels), were identified. The mean number of somatic mutations was 134 (range 28–693) per sample, or 2.89 (range 0.67–18.76) per Mb, and the mean number of indels was eight (range 3–228) (Appendix A). The 3947 somatic mutations and indels identified by whole-exome sequencing of 19 intestinal-type non-invasive GC samples contained 2658 non-synonymous mutations and 720 indels in 3078 genes (Appendix A). The most frequently occurring variant was C > T across the coding regions (Appendix A).

We found 50 genes that were mutated in more than three patients and had a mutation rate of >10 mutation/Mb. The most frequently mutated gene was *ERBB3* (mutated in 31.6% of our cohort), followed by *TP53* (26.3%), *BSN* (26.3%), *CSMD1* (26.3%), and *ADAMTS13* (26.3%) (Figure 2, Appendix A). However, no gene achieved experiment-wide significance following *p*-value adjustment (*q*-value < 0.05). In addition, we searched for mutation signatures in the exome-sequencing data. We found six samples that were strongly associated with signature six (Appendix A) and these were found to be associated with MSI (https://cancer.sanger.ac.uk/cosmic/signatures). Although no gene achieved the adjusted *p*-vale threshold (*q*-value < 0.05), we selected 50 recurrently mutated genes (mutated in more than three cases with a mutation rate of >10 mutation/Mb) in 19 samples for validation with Sanger sequencing.

#### 2.1.3. The Analysis of Significantly Associated Genes for Different-Type Non-Invasive GC

We performed a target panel which focused on 168 genes, combining 50 genes identified during whole-exome sequencing with 118 genes previously found to be frequently mutated in GC [11,12,13,14] (Appendix A). Then gDNA from 30 Japanese GC tissues were sequenced against the target panel. We achieved a 372-fold mean coverage for non-invasive GC (Appendix A). About 90% of the target regions achieved coverage of at least 30 reads, and a median of four somatic single-nucleotide variants (SNVs: range 0–26) and indels (range 0–17) were identified per sample (Figure 2a, Appendix A). We identified 56 genes that were mutated in at least four patients and had a mutation rate of > 10 mutation/ Mb. The most commonly mutated genes were *LRP1* (36.7%), *MLL2* (33.3%), *FAT1* (30.0%), *FAT4* (26.7%), *TP53* (23.3%), *ASH1L* (23.3%), *PCNT* (23.3%), and *MLL3* (23.3%) (Figure 2, Appendix A).

Using statistical analysis to define recurrent mutations in the target sequencing data, we identified two genes (*TP53*, *LRP1*) in which mutations were significantly over-represented (Appendix A). LRP1 (Lipoprotein receptor related-protein 1) is a member of the LDLR family protein and ubiquitously expressed in multiple tissues.

Then, we performed target sequence analysis of a total of six genes, *AID1A*, *APC*, *ERBB2* and *ERBB3*, in addition to *TP53* and *LRP1*, from gDNA extracted from cancerous and non-cancerous tissue from 19 advanced GC samples. We achieved a mean coverage of 188-fold for the cancerous region and 132-fold for normal region (Appendix A). About 90% of the target regions were covered by more than 30 reads. Among the six genes examined, the mutation rates were *TP53* (42.1%), *ERBB2* (31.6%), *LRP1* (21.1%), *APC* (29.0.%), *ARID1A* (10.5%), *ERBB3* (5.3%) (Appendix A).

The distribution of mutations in *LRP1* indicated that mutations tended to occur diffusely in some domains (Appendix A).

The rates of each representative GC driver gene identified in our study during the screening and replication stage were compared against ICGC (International Cancer Genome Consortium) data for Japanese patients. *TP53, CDH1, PIK3CA, SMAD4* had significantly higher mutation rates in ICGC the data than in our non-invasive GC data. *LRP1, APC,* and *ERBB3* had significantly higher gene mutation rates in our non-invasive GC data than in the ICGC data (Appendix A).

#### 2.1.4. Analysis of Expression of LRP1 and Accumulation of CagA

LRP1 is known to act as a suppressor or promoter of cancer [15,16,17]. In addition, it has already been reported that LRP1 is related to the intracellular autophagy of CagA, which affects the carcinogenesis of GC. CagA is a toxin that *Hp* injects into gastric epithelial cells and has been reported to be degraded through autophagy induced by binding of LRP1 and VacA, which is also an *Hp* exotoxin. This autophagy is induced via the mTOR pathway [15]. In the cancerous tissues in the cases for which FF samples were available, the expression level of LRP1 was examined based on the presence or absence of LRP1 mutations. There were no significant differences in LRP1 expression levels in the seven cancer cases with LRP1 mutations and the 18 cancer cases without LRP1 mutations (*p* = 0.809, Appendix A). Then we followed an in silico approach to predict the role of LRP1 mutation. We downloaded the RNA sequence data for 343 cases of gastric cancer (with/without LRP1 mutation: 26/327) from the TCGA Genomic Data Commons Data Portal and examined the differences in the LRP1 gene expression levels. LRP1 mRNA levels in gastric cancer with LRP1 mutation were significantly lower than those without LRP1 mutation in the TCGA–Stomach Adenocarcinoma (STAD) cohort (*p* < 0.001, Figure 3a). To gain further insight into the biologic pathways involved in gastric cancer stratified by the LRP1 mutation status, gene set enrichment analysis (GSEA) was performed in those cohorts. Enrichment plots of GSEA showed that the MTORC1 signaling gene set was more enriched in GC cases with LRP1 mutation, than GC cases without LRP1 mutation (NES = 2.8, FDR < 0.001, Figure 3b). Therefore, we performed fluorescent immunostaining to monitor the accumulation of CagA in a total of 49 cases (14 cases with LRP1 mutation and 35 cases without LRP1 mutation). Confocal microscope images of fluorescent immunostaining of CagA are shown in Figure 4. The accumulation of CagA was observed in 71.4% (10/14) of the cases with LRP1 mutation and in 22.9% (8/35) of the cases without mutation, but was significantly higher in cases with the LRP1 mutation (*p* < 0.05, Appendix A). Furthermore, even if we consider only 33 cases with HP infection (nine cases with LRP1 mutation and 24 cases without LRP1 mutation), CagA was observed in 100% (nine out of nine) of the cases with LRP1 mutation and in 25% (6/24) of the cases without mutation, which was significantly higher in cases with the LRP1 mutation (*p* < 0.05, Appendix A).

## 3. Discussion

We performed targeted sequencing of 19 non-invasive differentiated-type GC cases using genes identified during screening of whole-exome sequences of another 30 non-invasive GC cases, and found TP53 and LRP1 as significant mutations. Mutations in these two genes occurred at high frequencies in 19 advanced-stage GCs, following non-invasive differentiated-type GCs. We focused on LRP1 because it had not been reported as a strongly associated gene in previous studies of advanced gastric cancer. Using in silico approaches to predict the role of LRP1 mutation, we found that LRP1 mRNA levels in gastric cancers harboring LRP1 mutations were significantly lower than in those without LRP1 mutations. The MTORC1 signaling gene set was also enriched in GC cases with LRP1 mutations. Furthermore, we examined the accumulation of CagA by fluorescence immunostaining in the cancerous tissue and found that CagA had significantly accumulated in cases with LRP1 mutation. Furthermore, CagA had significantly accumulated in cases with mutations in LRP1, even when limited to cases currently infected with *Hp*. Two cases had CagA accumulation, despite eradication of *Hp* and no LRP1 mutations. In this study, only mutations in the present in exons were analyzed, and we speculate that there might have been a mutation in the splicing area.

In 49 non-invasive GC cases used for whole-exome sequencing and targeted sequencing, 14 samples (including two samples positive for EBV) had no major genetic mutations associated with GC, e.g., *TP53*, *ARID1A*, *PIK3CA*, *CDH1*, *ROHA* (Figure 2). There was a report that hypermethylation of CpG islands located in the promoter regions of tumor suppressor genes is an important mechanism for gene inactivation. Epigenetic alterations can cause a known genetic lesion in genes that are of key importance in the development of cancer. Some of the tumor suppressor genes are more frequently inactivated by hypermethylation than by genetic alterations in GC [18]. Therefore, DNA hypermethylation might play a role in such samples.

In this study, as a result of target sequence, LRP1 was identified as a significant mutation in addition to TP53. LRP1 is a 600kDa type I transmembrane receptor, located on chromosome 12, and consists of 84,844 bases and 89 exons. LRP1 is a member of the LDL-receptor gene family [19]. LRP1 has both endocytic and signaling activities. As a matricellular receptor, it is involved in regulation, mostly by clearing various extracellular matrix-degrading enzymes, including matrix metalloproteinases, serine proteases, protease inhibitor complexes, and the endoglycosidase heparanase. Furthermore, by binding extracellular ligands including growth factors and subsequent intracellular interaction with scaffolding and adaptor proteins, it is involved in the regulation of various signaling cascades [20]. Additionally, a role in cancer has been attributed to LRP1. LRP1 expression levels are often downregulated in cancer and some studies consider low LRP1 expression levels to be a poor prognostic factor [21].

Fluorescent immunostaining showed that CagA had significantly accumulated in the cancerous tissues of cases harboring LRP1 mutations. CagA is a carcinogenic protein injected into gastric epithelial cells by *Hp*. CagA that has invaded gastric epithelial cells induces aberrant cell proliferation signals and induces the polarity destruction of epithelial cells through the interaction with the oncoprotein SHP2 and the polarity control protein PAR1 [22,23]. However, the half-life of CagA in gastric epithelial cells is about 200 min, and proteolysis, depending on time, has also been reported [24,25]. Furthermore, the induction of autophagy *Hp*-infected cells has also been reported [26]. VacA, another toxin secreted by *Hp* has also been reported to induce autophagy in gastric epithelial cells [27]. The receptor for VacA-mediated induction of autophagy is LRP1 [15]. CagA is injected into epithelial cells by the type IV secretion system, but the binding of VacA and LRP1 induces autophagy, and CagA is degraded (Figure 5a). In the event of a decrease in LRP1 expression level or loss of function, autophagy is not induced, resulting in accumulation of CagA. VacA is roughly classified into m1-type VacA and m2-type VacA based on differences in gene sequence. While m1-type VacA showed induction activity against CagA-degrading autophagy, m2-type VacA showed no induction activity. Furthermore, CagA-degrading autophagy was not induced in cells in which LRP1 was knocked down, and CagA accumulated. It has also been shown that m2-type VacA, which does not show autophagy-inducing activity, has no ability to bind LRP1. Thus, the binding of m1-type VacA to LRP1 was considered to be an important initial response in the induction of CagA degradable autophagy [25]. In reports on the relationship between VacA and GC, cases of *Hp* infection in the stomach that led to GC had significantly more m1-type VacA. Looking at the global distribution of m1-type VacA, the proportion of m1-type VacA was relatively high in Central and South America, and Portugal and Spain. However, there was no difference in the distribution of m1-type VacA and m2-type VacA in North America, France, Italy, Northern and Eastern Europe, Australia, and East Asia (China, Hong Kong, Japan, and Thailand) [28]. On the other hand, the distribution of VacA in Japan is reported to be 97.7% (85/87) for m1-type VacA, which is clearly greater than in other countries and regions. Hypothetically, the low genetic heterogeneity in *Hp* vacA in Japan may be due to the traits of the country and people. Japan is a geographically isolated country with a relatively homogeneous population due to the fact that there has been very little mixing with other ethnic groups throughout its history [29]. In this study, 28.6% (14/49) of non-invasive GC cases and 31.0% (18/58) of both non-invasive and advanced GC cases harbored LRP1 mutations. Although the isolation of *Hp* and the examination of VacA have not been performed, there may be an association between the fact that LRP1 mutations were frequent in this study and that most *Hp* in Japan have m1-type VacA.

Previously, some groups reported the genomic landscape of non-invasive differentiated-type GCs by target-sequencing or analysis of the microsatellite instability (MSI) status, copy number aberrations (CNAs), and single-nucleotide polymorphisms (SNPs) [30,31]. Rokutan et al. performed target sequencing on 68 genes in 18 non-invasive GC cases and concluded that TP53 mutation is the first event in the progression of non-invasive GC cases with TP53 mutation. In addition, they reported that TP53 mutation was absent in MSI non-invasive GCs (0/4) [14]. Even in our study, only one case out of 10 MSI cases had a TP53 mutation. Moreover, Yoshida et al. performed target sequencing on 50 genes for 31 non-invasive GCs. They reported that mutation in TP53 and Wnt-signaling pathway genes (APC and CTNNB1) were frequently found and were mutually exclusive [32]. This result is almost the same in our data, that is, 10 out of 12 cases with TP53 mutation had no mutation in APC and CTNNB1. Furthermore, the frequency of the following gene mutations did not differ significantly between our data and their data (*ARID1A, ARID2, APC, SMAD4*, *PIK3CA*, *RHOA*, *KRAS*, *MUC6*, and *APC*: Fisher’s exact test, data not shown) [14,32]. However, LRP1, which was detected as a significant gene mutation following TP53 in our study was not included in their target-sequencing panel.

Mutations of ARID1A and PIK3CA are known to be mutually exclusive in GC, with similar results in this study. The present results suggested that ARID1A and PIK3CA may be associated in the development and progression of GC via a specific mechanism. In particular, the PIK3CA mutation rate has been reported to be associated with GC penetration [16], and ICGC reported mutation rates of 9.7% in Japanese patients. However, in this study there was no case of PIK3CA mutation. This seems to be due to the fact that all target cases in this study were intramucosal lesions.

This study has several limitations. The most crucial limitation was that the sample size was less than 100, which limits the possibility of detecting all possible changes in cancer driver genes. In previous studies, *CDH1*, *SMAD4*, *PIK3CA*, *RHOA*, *ARID1A*, *KRAS*, *MUC6*, and *APC* were significantly associated with the development of GCs, but we did not find corroborating evidence in support of each gene in the case of non-invasive differentiated-type GCs in this study, which might be due to sample size. Secondly, this hypothesis accounts for only 28.6% (14/49) of non-invasive diffuse-type GC cases. Other mechanisms may operate on the remaining cases. Third, the association between mutations and the accumulation of CagA by immunohistochemistry is not conclusive evidence that somatic mutation of LRP1 is sufficient to drive diffuse-type GC development from *Hp*-infective gastric mucosa. Further studies of LRP1 mutations, especially in premalignant mucosa, will be required to clarify the relationship between chronic inflammation by *Hp* infection, somatic mutation of LRP1, and diffuse-type GC development.

## 4. Materials and Methods

### 4.1. Sample Collection

This research was approved by the Hiroshima University Human Genome Ethical Committee. Informed consent for whole-genome analysis was obtained from all patients. All samples were obtained with the approval of the ethics committee of Hiroshima University Hospital. The approval code from the ethical committee is “ヒ-90” and the date is “15 October 2012”.

A total of 49 distinct samples from 49 individuals with non-invasive differentiated-type GCs and a total of 19 distinct samples from 19 individuals with advanced GCs were used in this study.

Samples of non-invasive differentiated-type GCs were collected from patients who underwent endoscopic submucosal dissection (ESD) between April 2012 and February 2016. Nineteen fresh-frozen (FF) biopsy samples were collected from 19 patients for whole-exome sequencing. We also used matched blood samples for comparison with the germline sequence. Additionally, we used 30 formalin-fixed paraffin-embedded (FFPE) samples of 30 other non-invasive GC patients.

Samples of advanced GC were collected from patients who underwent surgery between April 2014 and December 2017. Nineteen paired FF samples (19 cancerous and 19 normal) for targeted sequencing were collected from surgical specimens from 19 patients.

FF biopsy samples were collected from cancerous tissue in both non-invasive GC patients and advanced GC patients. We used 10μm thick unstained FFPE tissue and isolated cancerous areas by microdissection with comparison to an H and E-stained slide identified by a pathologist. All background gastric mucosa had the features of chronic gastritis and mucosal atrophy with *Hp* infection, and none of the gastric specimens included dysplastic lesions.

### 4.2. Evaluation of Hp Infection

Before ESD, the presence or absence of *Hp* infection was evaluated by a urea breath test (UBT) or anti-*Hp* IgG antibody. If there was diffuse redness or atrophy in the epithelium of the background gastric mucosa in the endoscopic image, it was categorized as a past or active infection.

### 4.3. Purification of DNA from Fresh Frozen and FFPE Tissue

DNA was extracted from frozen samples, FFPE specimens, and blood leukocyte samples using the ALLPrep^®^ DNA/RNA Micro Kit, GeneRead^TM^ DNA FFPE Kit, and QIAamp DNA Blood Midi/Maxi kit, respectively (Qiagen, Hilden, Germany). DNA volume was measured by Qubit HS (Qiagen). The quantity and quality of the FFPE-derived DNA was assessed by calculating normalized DNA integrity scores (ΔΔCq) using quantitative PCR with the Agilent NGS FFPE QC Kit (Agilent Technologies, San Diego, CA, USA). Genomic DNA > 60 ng with high quality was selected for library preparation.

### 4.4. Preparation and Sequencing of DNA Exome-Seq and Target-Seq Libraries

Genomic DNA (gDNA) libraries were prepared from FF-derived DNA using the Agilent Sure Select^xt^ Target Enrichment System Automation Protocol (Agilent Technologies, Tokyo, Japan), and gDNA library preparation was done according to the manufacturer’s instructions, using 200ng DNA. gDNA libraries were prepared from FFPE-derived DNA using Agilent Haloplex^xt^ Target Enrichment System Automation Protocol (Agilent Technologies, Tokyo, Japan). The size distributions of the gDNA libraries were estimated by on-chip electrophoresis (High-Sensitivity DNA chips and High-Sensitivity D1000 Screen Tape) of a 1 µL sample on an Agilent 2100 Bioanalyzer/Agilent 2200 Tapestation (Agilent Technologies, Tokyo, Japan). The exome or target genes captured gDNA libraries were combined into 2 nM pooled stocks, denatured and diluted to 10 pM with pre-chilled hybridization buffer and loaded into TruSeq PE v3 flowcells on an Illumina cBot, followed by indexed paired-end sequencing (101 + 7 + 101 bp) on a Illumina HiSeq 2500 using TruSeq SBS Kit v3 chemistry (Illumina, San Diego, California, USA).

### 4.5. Sequence Analysis

Paired de-multiplexed FASTQ files were generated using the CASAVA software (Illumina), and initial quality control was also performed using CASAVA. Paired de-multiplexed FASTQ files from DNA-exome libraries were imported into the CLC Genomics Workbench (CLC Bio, version 8.5.1) running on CLC Genomics Server (CLC Bio, version 5.0.1). The analysis of non-invasive GCs used the CLC 8.5.1 version Genomics Workbench (CLC bio, Aarhus, Denmark). Sequences were aligned to the hg19/GRCh37 reference sequence and analyzed using the Map Reads to Reference tool with default parameters to generate a binary sequence alignment map (BAM) file. The aligned BAM file was sorted and merged using the Merge Read Mappings tool. PCR duplicates were removed using the Removed Duplicate Mapped Reads tool, and local realignment was performed using the Local Realignment tool to improve mapping quality prior to screening for variants. To identify variants, the low-frequency variant detection tool (http://resources.qiagenbioinformatics.com/manuals/clcgenomicsworkbench/current/User_Manual.pdf, p653–6) was used. We set the following criteria for identification of reliable somatic SNVs or indels: (1) read depth over the mutated site should be at least 30, with at least three reads harboring the mutation; (2) the mutant allele should be present in at least 10% of all reads; (3) read depth covering the mutated site in the corresponding normal control should also be at least 30, with, at most, one read harboring the mutation; (4) variants, for which the minimum of the fraction of ‘countable’ forward reads and ‘countable’ reverse reads carrying the variant to all ‘countable’ reads carrying the variant is less than 0.2, are excluded; and (5) mutations listed in dbSNP 137, the HapMap database, or the 1000 Genomes Project are excluded.

In the analysis of advanced-stage GC, the trimmed paired-end reads were mapped to the human reference genome (UCSC Human Genome Reference hg19), and the mutation detection process was performed using the SureCall System 4.0.1.46 Haloplex Default Analysis Method (Agilent Technologies). Variant calling was performed by SNPPET in the SureCall System 4.0.1.46, and the criteria for the identification of reliable somatic SNVs or indels were set as follows: (1) variant score threshold of 0.3, minimum quality for base 30; (2) variant call quality threshold of 100; (3) minimum allele frequency of 0.1; (4) minimum number of reads supporting variant allele of 10; (5) minimum number of read pairs per barcode of 2; and (6) removal of mutations listed in dbSNP 137, the HapMap database, or the 1000 Genomes Project.

### 4.6. Mutation Validation by Sanger Sequencing

To validate the results of exome sequencing for non-invasive GC in the screening stage, we confirmed a total of 134 putative somatic mutations identified by exome sequencing by independent PCR and Sanger sequencing (Appendix A). In all cases, the somatic origin of the mutation found in the tumor was verified by sequencing the corresponding peripheral blood leukocyte samples.

### 4.7. Epstein-Barr Virus (EBV)

We used Epstein-Barr virus-encoded small RNA (EBER) in situ hybridization (Leica Biosystems, Buffalo Grove, IL, USA) to detect EBV-positive GC.

### 4.8. MSI Analyses

Microsatellite loci (BAT25, BAT26, D2S123, D5S346, and D17S250) were amplified according to recommendations by the Bethesda guidelines [33]. DNA extracts (two of 50 uL) were assessed via the Multiplex-PCR approach (Toyobo, Osaka, Japan) according to the manufacturer instructions, using an annealing Tm of 60 °C. The amplification of BAT25 and D2S123 loci and that of D5S346 and D17S250 loci were combined in the duplex assays. To separate the microsatellite PCR products, the DNA 1000 LabChip Kit and Agilent 2100 Bioanalyzer were utilized according to the instructions provided by the manufacturer. The fragment analysis was carried out using the Agilent 2100 Expert software (Agilent). To identify MSI in the non-invasive GC cases, an overlay of two electropherograms was used to compare the PCR patterns as previously described [34]. The MSI levels were classified into three levels based on the Bethesda guidelines [33]: MSI-high (MSI-H) is generally defined as MSI in more than 30% of the standard markers; MSI-low (MSI-L) is generally defined when changes are exhibited in less than 30% of the markers; and microsatellite stability (MSS) is generally defined in the absence of microsatellite alterations.

### 4.9. Analysis of Mutation Signature

We made a Python script to extract each Excel sheet into a separate file containing four columns (CHROM, POS, REF, ALT) and uploaded the files to the http://bioinfo.ciberehd.org:3838/MuSiCa/ website as TSV files with exome sequencing against the hg19 reference sequence [35]. Information on mutation signatures was obtained from the COSMIC website (https://cancer.sanger.ac.uk/cosmic/signatures).

### 4.10. Identification of Significantly Mutated Genes

Because the number of mutations in a gene is influenced by the gene length and the background mutation rate, we calculated the probability of the number of protein-altering mutations under the given mutation rate and gene length using the following set of calculations. First, we divided the genomic region into 1-Megabase (Mb) bins and estimated the mutation rates for point mutations and indels. Because the mutation rates in CpG sites were much higher than those of other regions, we estimated the mutation rate for point mutations in CpG and non-CpG sites separately. We used mutations in non-coding regions for mutation rate estimation. Second, the number of nonsynonymous and synonymous sites was counted for each gene. Finally, the expected number of mutations in each gene was calculated by the total number of nonsynonymous and synonymous sites and the background mutation rate. Tests of significance for each gene were performed using Fisher’s exact test. We adjusted for multiple testing using the Benjamini–Hochberg method [36].

### 4.11. LRP1 mRNA Analyses

RNA was reverse-transcribed using QuantiTect^®^ Reverse Transcription Kit (QIAGEN, Hilden, Germany) according to the manufacturer’s instructions. Real-time PCR was performed using an Absolute SYBR Green mix (Agilent Technologies, Tokyo, Japan), on Mx3000P system (Agilent Technologies, Tokyo, Japan) using LRP1 primers (5′-CTG CTC TCA GCT CTG GTC G-3′ and 5′-AAA TCT CAG GGG CCT CGT CA-5′). LRP1 primers were synthesized by Sigma-Aldrich (Sigma Aldrich Japan LLC, Tokyo). PCR conditions were set at 10 min at 95 °C, followed by 40 cycles, each consisting of 30 sec at 95 °C (denaturation), 1 min at 50 °C (annealing) and 1 min at 72 °C (extension). The specificity of PCR amplification was checked using a heat dissociation curve from 55 °C to 95 °C following the final cycle. The cycle threshold (Ct) values were recorded with MxPro^TM^ QPCR software (Agilent Technologies, Tokyo, Japan). The results are expressed as the mean of experiments performed in triplicate.

### 4.12. Bioinformatics Analysis

TCGA RNA-Seq data and the information concerning the LRP1 mutation obtained from 343 gastric cancers in the The Cancer Genome Atlas Stomach Adenocarcinoma (TCGA–STAD) were downloaded from the TCGA website (https://tcga-data.nci.nih.gov/tcga/).

### 4.13. Gene Set Enrichment Analysis

Gene set enrichment analysis (GSEA) was performed as described [37] to analyze the differential modulation of molecular pathways in the Hallmark gene set (http://software.broadinstitute.org/gsea/msigdb/collections.jsp) using the GenePattern Survival Analysis module (www.broadinstitute.org/genepattern).

### 4.14. Immunofluorescence Staining of CagA

Paraffin-embedded human non-invasive GC tissue was cut into 2–3 μm sections and mounted on positively charged slides. Antigen retrieval was conducted with citrate pH6 buffer in a microwave oven at 800 W for 5 min and at 150 W for 10 min. The slides were then incubated with a primary antibody overnight at 4 °C. The primary antibody was produced from a mouse monoclonal antibody against amino acids 1–300 of CagA of *Hp* origin. (Santa Cruz Biotechnology, Santa Cruz, CA, USA). The fluorescent signal of the secondary antibody was captured using confocal laser-scanning microscopy (Carl Zeiss Microscopy, Thornwood, NY, USA).

## 5. Conclusions

This study provides a detailed profile of somatic mutations in 49 non-invasive diffused type GCs. The profile revealed that *LRP1* was mutated and functioned as a cancer driver in about 28.6% of cases. Somatic mutations of LRP1 were associated with the accumulation of CagA (Figure 5b). These findings may be helpful to clarify the driving mechanism for intestinal-type GCs. Although it is necessary to validate prospective trials or large-scale clinical trials, we think that our findings could lead to the development of new molecular imaging for gastric cancer.

## Figures and Tables

**Figure 1 cancers-12-00510-f001:**
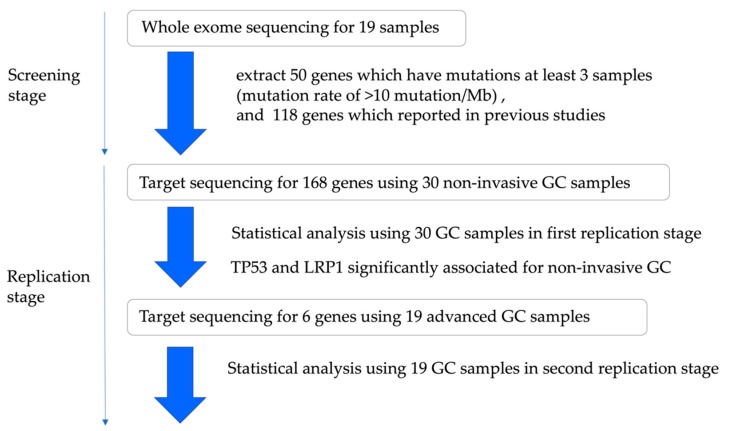
Outline of the study design. We performed whole-exome sequence of 30 non-invasive gastric cancer (GC cases) and, using Sanger sequencing, we were able to validate 19 cases. There were 50 genes which were mutated in more than three patients and had a mutation rate of >10 mutation/Mb. As a replication study, deep sequence was performed in another 30 non-invasive GC cases for 168 genes, including the 50 genes and 118 previously reported gene mutations. Here, gene mutations of TP53 and LRP1 have been identified as significant in non-invasive cancer. In addition, deep sequence was performed in 19 advanced GC cases for six genes, including TP53 and LRP1.

**Figure 2 cancers-12-00510-f002:**
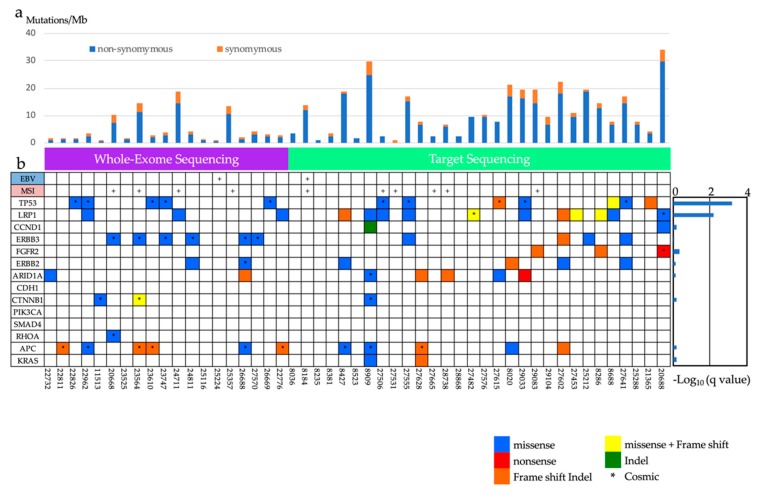
Summary of frequently mutated genes in 49 non-invasive GCs. (**a**) Bars represent the number of somatic mutations (somatic single-nucleotide variations (SNVs); small insertions and deletions (indels)) with synonymous and non-synonymous mutation rates distinguished by color. (**b**) Frequently mutated genes are represented by each sample. Epstein-Barr Virus (EBV)-positive cases and microsatellite instability (MSI)-positive cases are indicated at the top of the graph. Mutation color indicates the class of mutation. Twenty-eight LRP1 mutations were observed in 15 cases. EBV, Epstein-Barr virus; MSI, microsatellite instability.

**Figure 3 cancers-12-00510-f003:**
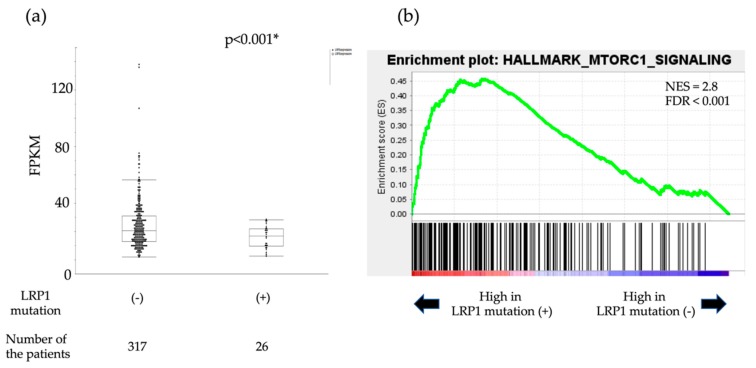
In silico approaches to predict the role of LRP1 mutation. (**a**) Boxplot representing the mRNA level in The Cancer Genome Atlas Stomach Adenocarcinoma (TCGA–STAD) cohort. Fragments per kilobase of exon per million (FPKM) mapped those in Figure 1 with mutation and those without mutation. (**b**) The enrichment plot, light green line, providing a graphical view of the enrichment score for MTORC1 signaling gene. The score at the furthest positive peak indicates the enrichment score for this gene set. The lower vertical lines show the location of genes from the gene set. Gene set enrichment analysis (GSEA); normalized enrichment score (NES); false discovery (FDR).

**Figure 4 cancers-12-00510-f004:**
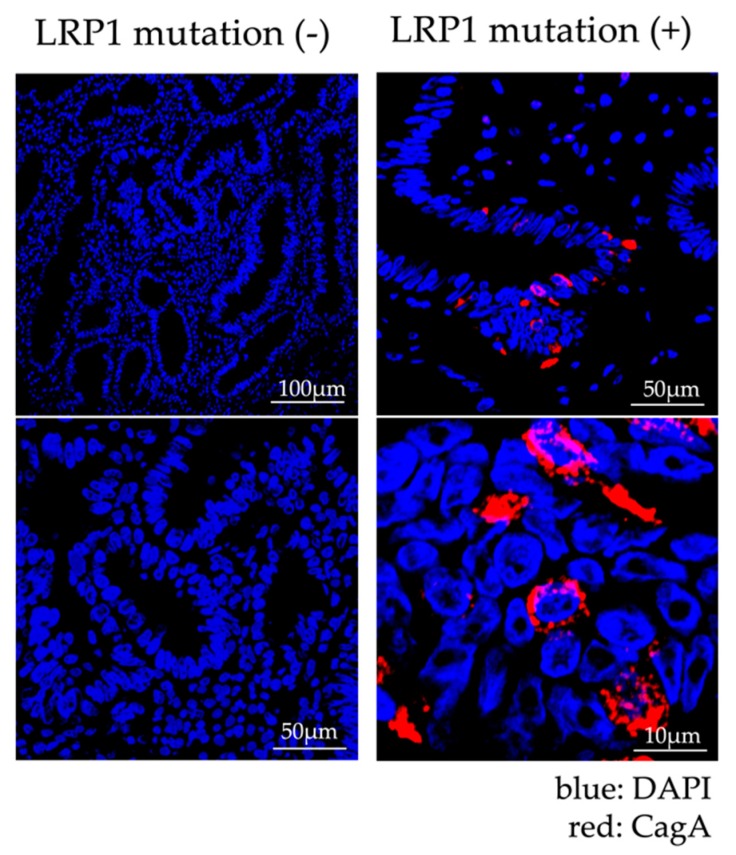
Immunofluorescence Staining of specimens treated by endoscopic submucosal dissection (ESD). The case with LRP1 mutation is shown on the right. CagA staining (red) was observed in the cytoplasm of gastric cancer cells in LRP1 mutant cases. On the other hand, the case without LRP1 mutation is shown on the left. CagA staining was not found in the cytoplasm of gastric cancer cells in cases without LRP1 mutation. DAPI (4’,6-diamidino-2-phenylindole) nuclear staining is shown in blue.

**Figure 5 cancers-12-00510-f005:**
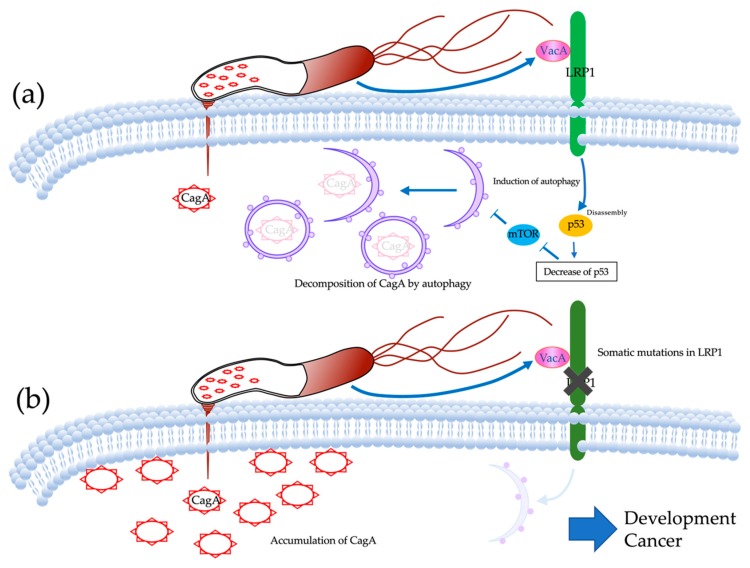
The mechanism of developing gastric cancers by LRP1 mutations. (**a**) CagA is injected into epithelial cells by the type IV secretion system, but binding of VacA and LRP1 induces autophagy and CagA is degraded. (**b**) LRP1 mutations damage the intake of VacA into gastric epithelial cells, inhibit autophagy of CagA, and then develop gastric cancers.

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
