# Peer review of "Genomic Characterization of Non-Invasive Differentiated-Type Gastric Cancer in the Japanese Population"

_cancers, 2020, doi:10.3390/cancers12020510_

Round 1

Reviewer 1 Report

Thanks to the authors for their valuable work.

I have some minor concerns:

Table 1 and 2 seem to be irrelevant. Can't you report data directly in the text and delete them? I feel there is no future perspective reported into your paper. Please improve the conclusion. What would be the clinical application of these results? Any prospective trial, bigger trials with higher number of patients?

Author Response

"Table 1 and 2 seem to be irrelevant. Can't you report data directly in the text and delete them? I feel there is no future perspective reported into your paper."

Thank you for the useful comments.

As reviewer1 point out, we deleted Table. Then we changed and added the manuscript as follows (page ï¼” lines 167- page5 lines 173);

From” CagA staining was found in the cytoplasm of GC cells, and CagA was significantly accumulated in cases with LRP1 mutations (p < 0.05, Table 1, Supplementary Table 13). Furthermore, CagA was significantly accumulated in cases with mutations in LRP1 even when limited to cases currently infected with Hp (p < 0.05, Table 2, Supplementary Table 13).”

To” The accumulation of CagA was observed in 71.4% (10/14) of the cases with LRP1 mutation and in 22.9% (8/35) of the cases without mutation, which was significantly higher in cases with the LRP1 mutation (p < 0.05, Table S13). Furthermore, even if we consider only 33 cases with HP infection(9 cases with LRP1 mutation and 24 cases without LRP1 mutation), CagA was observed in 100% (9/9) of the cases with LRP1 mutation and in 25% (6/24) of the cases without mutation, which was significantly higher in cases with the LRP1 mutation (p < 0.05, Table S13).”

"Please improve the conclusion. What would be the clinical application of these results? Any prospective trial, bigger trials with higher number of patients?"

Thank you for your valuable comments.

As reviewer1 point out, we added conclusion as follow (page 15, line 467-9);

Although, it is necessary to validate prospective trial or large-scale clinical trials, we think that our finding would lead to the development of new molecular imaging of gastric cancer.

Reviewer 2 Report

Thank you for the opportunity to revise this interesting paper. I think that it is very well written and documented, so I strongly recommend its publication in the present form. 

Author Response

Thank you for review. I am very honored that you have strongly recommended publication. I will edit while valuing your review.

Reviewer 3 Report

Overall, the manuscript was well written and worthy of publiction.

However, before publication, the quality of image should be improved. 

Ex) Fig 1. figure caption was indiscernible.

Full name of LRP1 was appeared last part of manuscript.

Shoule be check the format of table.

Author Response

"Overall, the manuscript was well written and worthy of publiction.

However, before publication, the quality of image should be improved.

Ex) Fig 1. figure caption was indiscernible.

Full name of LRP1 was appeared last part of manuscript.

Shoule be check the format of table."

Thank you for the useful comments.

According to reviewer comment, we changed and added the manuscript as follows (page ï¼” lines 130-131);

From” Using statistical analysis to define recurrent mutations in the target sequencing data, and identified 2 genes (TP53, LRP1) in which mutations were significantly over-represented (Supplementary Table 10).”

To” Using statistical analysis to define recurrent mutations in the target sequencing data, and identified 2 genes (TP53, LRP1) in which mutations were significantly over-represented (Table S10). LRP1 (Lipoprotein receptor related-protein 1) is a member of the LDLR family protein and ubiquitously expressed in multiple tissues.”

Moreover, we changed that from “LRP1 (Lipoprotein receptor related-protein 1), a 600kDa type I transmembrane receptor, is located on chromosome 12 and consists of 84,844 bases and 89 exons.” to “LRP1 is a 600kDa type I transmembrane receptor, located on chromosome 12, and consists of 84,844 bases and 89 exons.” in page 18 line 242-3.
